# Research on Construction and Application of Ocean Circulation Spatial–Temporal Ontology

Hao Zhang [1], Anmin Zhang [1,2,*], Chenxu Wang [1,*], Liuyang Zhang [3] and Shuai Liu [1]

1   School of Marine Science and Technology, Tianjin University, Tianjin 300072, China; haozhang86@tju.edu.cn (H.Z.); liu245328495@tju.edu.cn (S.L.)
2   Tianjin Port Environmental Monitoring Engineering Technology Center, Tianjin 300072, China
3   State Key Laboratory for Manufacturing Systems Engineering, Xi'an Jiaotong University, Xi'an 710049, China; liuyangzhang@xjtu.edu.cn
*   Correspondence: anmin.zhang@tju.edu.cn (A.Z.); chenxu.wang@tju.edu.cn (C.W.)

**Abstract:** Due to the absence of a comprehensive knowledge system for modeling ocean circulation, there is ambiguity and diversity in the semantic expression of ocean circulation. This makes it difficult to organize and share relevant spatiotemporal data effectively. This paper addresses the issue of ocean circulation by introducing ontological theory and methodology based on a comprehensive analysis of domain knowledge. Through a comprehensive analysis of the conceptual and relational characteristics of different elements, we define classes, properties, spatiotemporal relationships, and inference conditions with which to formally express concepts and relationships in ocean circulation, and finally complete the construction of ocean circulation ontology. The formal expression of the Equatorial Counter Current is presented as an example with which to validate the effectiveness of ontological construction. Additionally, an ontology-based knowledge base of ocean circulation is proposed. The construction framework is described, and several examples of knowledge base queries are also illustrated. The results demonstrate that this ontology can effectively represent the relevant knowledge within ocean circulation and provide a meaningful reference for investigating knowledge sharing and semantic integration within this field.

**Keywords:** ontology; spatial–temporal ontology; ocean circulation; semantic analysis; Web Ontology Language (OWL)

## 1. Introduction

Ocean circulation plays a crucial role in the exchange of matter and energy within the ocean. It exerts a significant influence on the distribution of marine life, air–sea interactions, and climate change. With the continuous development of oceanographic observation technologies, humans have gained access to a vast amount of oceanographic data and information. However, because of the diversity of data sources and representations, there are several issues, such as inconsistent data storage formats and unclear conceptual classification, as well as synonyms and polysemy. While metadata ensure consistency at the syntactic level, they cannot resolve semantic ambiguity. This semantic heterogeneity poses great difficulties for knowledge sharing and the integration of ocean data. Therefore, it is crucial for ocean science research, as well as for data organization, integration, and sharing, to establish a formalized expression system that better reflects the knowledge of ocean circulation and achieves a unified description of ocean circulation at the semantic level.

Ontology is defined as a hierarchical and relational set of vocabulary that is used to describe concepts within a specific domain and the relationships between them [1]. An ontology can capture domain-specific knowledge, determine common vocabulary terms, and establish their interrelationships across different levels of formal patterns within the domain. Therefore, the introduction of ontology into ocean science research can effectively address the aforementioned problems.

At present, in the realm of developing knowledge systems for marine phenomena, Bermudez et al. [2] proposed an ontology for marine platforms to delineate practical ontological design principles and presented important platform terminology for the European marine community. However, the specific interactions between concepts have not been described and expressed. Tao et al. [3] utilized the National Oceanography Centre Southampton's Ferrybox project to establish an ontology-based reference model of a Collaborative Ocean, where relevant oceanographic resources can be semantically annotated to produce Resource Description Framework (RDF)-format metadata in order to facilitate data/resource interoperability in a distributed environment. Xiong et al. [4] proposed marine ecology ontologies based on role theory to solve the data sharing and knowledge representation problems of marine scientific research. Yun [5] proposed a knowledge engineering approach and applied it to construct an ontology for marine biology. The Biological and Chemical Oceanography Data Management Office (BCO-DMO) has developed an ontology project aimed at enhancing the interoperability of ocean biogeochemical data [6]. Abidi et al. [7] presented a knowledge modeling solution for ocean knowledge and data management. Jia et al. [8] proposed an ontology model using the Hozo role theory. They then built an ontology around the ocean carbon cycle to describe and share the fundamental knowledge of the cycle. In 2017, Stocker [9] conducted research on the semantic representation of marine monitoring data. The study demonstrated the application of a technological framework in experiments that integrated sensing data and metadata from heterogeneous and distributed resources.

Currently, research on ocean circulation ontology includes the Marine Metadata Interoperability (MMI) [10], the Semantic Web for Earth and Environmental Terminology (SWEET) [11,12], and the Arctic Report Card (ARCRC) [13]. Both the MMI and SWEET projects offer descriptions of the majority of fundamental concepts within their marine science domains. However, they do not provide specific descriptions of the interrelationships among concepts in ocean circulation. The ARCRC project focuses on describing the Arctic domain but lacks a comprehensive description of ocean circulation knowledge. Li [14] proposed a quadruple-based organization for ocean circulation features and performed a collaborative filtering method with which to assess the similarity of structural features in ocean circulation. This approach provides semantic support for the integration and management of spatiotemporal data in ocean circulation. Ji et al. [15] constructed a three-tier architecture to achieve the semantic expression of ocean current domain knowledge. Based on the syntactic rules of the Web Ontology Language (OWL), the researchers constructed spatiotemporal predicates and terms, extended OWL spatial–temporal modeling key nodes, and developed the Ocean Current Spatiotemporal Web Ontology Language (OST OWL). However, the methods they proposed also lack specific descriptions of the relationships between concepts in ocean circulation.

Presently, there is still an absence of a comprehensive description of the knowledge system about ocean circulation. Therefore, it is necessary to comprehensively analyze the internal characteristics and deeply explore the conceptual relationships within the field of ocean circulation. Building upon existing ontological semantic models and methodologies, this paper presents an ontological modeling and knowledge representation system for ocean circulation. The main contributions of our work are outlined below:

(1) The formal description of ocean circulation based on an analysis of spatial–temporal semantic relationships.

(2) An ontology-based knowledge base for ocean circulation is developed to facilitate the sharing, reuse, and reasoning of implicit knowledge.

The remainder of this paper is structured in the following manner: Section 2 analyzes the features of ocean circulation. Section 3 focuses on constructing the ontology of ocean circulation. Section 4 presents a framework for an ontology-based knowledge base of ocean circulation. In Section 5, the results are analyzed and discussed.

## 2. Feature Analysis of Ocean Circulation

The study of ocean phenomena relies on the availability of ocean data resources obtained through field observations, numerical simulations, and satellite remote sensing. These data resources provide a foundation for analyzing and calculating the structure and characteristics of ocean currents. They allow us to uncover various patterns and laws underlying ocean phenomena. The types of oceanic data related to the study of ocean currents are diverse and include observational data, remote sensing data, and numerical simulation data.

Ocean circulation is the fundamental element of the ocean system and has significant impacts on marine ecosystems and material exchange. The complexity of ocean current movement determines the diversity of its characteristics [16,17].

Ocean circulation exhibits multidimensional characteristics, including diversities in attribute dimensions, spatial topological features, and fuzzy features [18]. The attribute information of ocean circulation is divided into three categories: vector attributes, scalar attributes, and physical quantity characteristic attributes. Vector attributes describe motion states (velocity, direction, and dispersion), while scalar attributes express the physical properties (temperature, salinity, and density) of currents. Oceanic circulation exhibits unique phenomena with distinct properties. When studying the vortex, scholars focus on attributes such as vorticity, vortex flux, and velocity circulation for characterization and extraction [19].

Ocean circulation typically exhibits spatial topological features with indistinct boundaries. Ocean currents form a continuous system, with various current phenomena interacting with each other to create distinct yet interconnected characteristic regions. In the topological analysis of ocean circulation, topology theory [20] is applied to extract critical points from ocean circulation data in order to reflect the spatial topological structure of ocean circulation. Furthermore, ocean circulation also exhibits fuzzy features, which are evident in the unclear boundaries. Unlike terrestrial objects, which have well-defined boundaries and can be precisely described through measurement techniques, ocean currents are constantly in motion, making it challenging to accurately delineate their boundaries. In flow field analysis, it is often necessary to use feature analysis, cluster analysis, and other methods to approximate the boundaries [21]. The Kuroshio is a relatively warm ocean current that has an average annual sea surface temperature of about 24 °C. It is approximately 100 km wide and generates frequent small to mesoscale eddies [22]. The usage of terms such as "relatively", "average", "about", "approximately", and "small" introduces ambiguity that can impact the consistency of semantic expression.

## 3. Ocean Circulation Domain Ontology Construction

### 3.1. Definition of the Ontological Structure of Ocean Circulation

The essence of ontology modeling lies in the accurate definition of spatial data concepts, including their attributes, constraint conditions, and hierarchical relationships within a specific domain. Currently, many researchers have proposed models, such as triples [23,24], quadruples [25], quintuples [26], and sextuples [27], to represent geographical ontologies. These models provide an effective way to integrate geographical concepts and relational rules within the geographic domain. Particularly, the triplet model is an abstract form of ontology modeling that encounters limitations in describing detailed ontological information. The quintuple model separates attributes and emphasizes their importance. The sextuple model provides a more detailed classification of attributes and relationships, but this increased specificity may limit its applicability. Based on the concept of geographical ontology, and incorporating its existing logical structure, it is believed that the quintuple model is sufficient to represent the ocean circulation ontology accurately.

$$O_{\text{ocean\_circulation}} = \{C(c_1, c_2, \ldots, c_n), R(r_1, r_2, \ldots, r_n), P(p_1, p_2, \ldots, p_n), A, I(i_1, i_2, \ldots, i_n)\} \quad (1)$$

Equation (1) outlines the different components of the proposed ontology structure for ocean circulation. The set of definitions for concepts related to ocean circulation is represented by the symbol '*C*'. '*R*' refers to various relationships within ocean circulation, including relationships

between different concepts, between concepts and attributes, between different attributes, and between concepts and instances. '*P*' represents the attribute characteristics of ocean circulation, while '*A*' conveys the recognized laws, knowledge, and rules or constraint conditions governing geospatial concepts. For instance, in the context of ocean circulation, it is widely accepted that spatial distance cannot be negative. Finally, '*I*' denotes the set of ocean circulation instances.

In our work, we propose our approach to construct an ontology model for ocean circulation based on Noy's seven-step method [28]. We consolidate several steps from the seven-step method, such as defining classes and the class hierarchy, defining the properties of classes–slots, and defining the facets of the slots, into a single step called the formal construction of the ontology. Furthermore, we have implemented additional steps in order to validate the effectiveness of the ontology [29,30], ensuring that it meets the requirements for knowledge representation within the considered domain. These improvements aim to enhance the reliability and rationality of ontology construction. Figure 1 shows the process of constructing ocean circulation ontology. The specific steps are as follows:

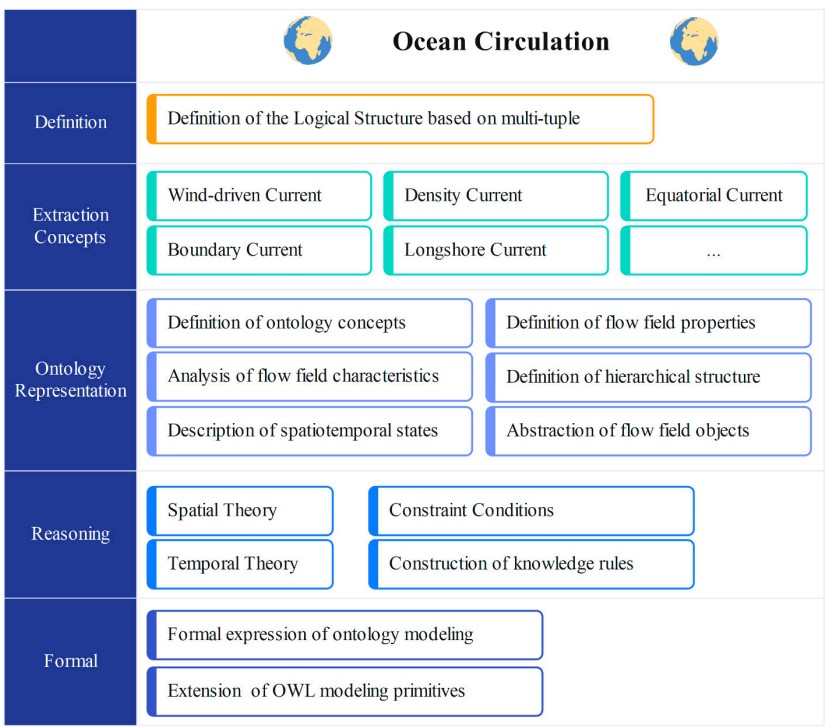

**Figure 1.** The process of constructing ocean circulation ontology.

Step 1: Determine the purpose and scope of ontology for ocean circulation, and clearly define the concepts, properties, and relationships within this domain to meet the requirements for the semantic level.

The domain of ocean circulation encompasses the intricate relationships among ocean currents, marine environments, oceanography, and marine economics. The formal representation of ocean circulation has been defined as a focused domain that specifically targets the definition and spatiotemporal processes associated with ocean circulation. The purpose of ontology development is to establish a computer-interpretable knowledge sharing and management system for ocean circulation. This system aims to provide a robust platform by which to support marine scientific research and foster collaboration within the field.

Step 2: Analyze the current concepts and descriptions of ocean circulation, utilize the pre-existing ontology to define relevant knowledge, repurpose the knowledge, and finalize the expression of the domain ontology and the creation of the knowledge base. This involves referencing marine science, international standards, professional literature, databases, research reports, and other relevant resources. Creating a knowledge base can clarify the specific concepts and interrelationships involved in ocean circulation, enabling the reuse of domain knowledge. Due to the complexity of the subject matter when constructing a knowledge base for ocean circulation, it is crucial to seek the assistance of marine experts, in order to ensure the accuracy and precision of the knowledge base and experimental data.

Step 3: Analyze the listed concepts and relationship systems, and define the concepts, relationships, properties, and constraint conditions within ocean circulation ontology.

In this step, it is necessary to extract the associated terminology and identify the unanimously approved domain concept. This requires establishing the level of conceptualization and determining a consistent naming convention. It is also crucial to examine the semantic relationships during the spatiotemporal development of ocean circulation. During this process, assistance from marine experts is essential to ensure precise descriptions and thorough analysis. Further elaboration on these topics can be found in Section 3.2.

Step 4: Formally describe the ontology of ocean circulation.

In order to finalize the formal representation of the ontology for ocean circulation, the OWL language is utilized in conjunction with the Protégé software platform to create concepts, properties, instances, and relationships. More information on this step can be found in Section 3.3.

Step 5: Validate the adequacy of the ontology in describing various data, and ensure the effectiveness of ocean circulation ontology.

The ontology undergoes expansion and enhancement based on the evaluation and analysis results. In practice, ontology construction typically follows a spiral approach that involves iterative and incremental processes designed for continual improvement.

Step 6: Create an instance of ocean circulation ontology.

In this step, ontology instances are actively created and associations are established with their corresponding entries in the database.

### 3.2. Analysis of Spatial–Temporal Process Semantics

Given the complexity of spatiotemporal processes in ocean circulation, we will leverage existing ontology knowledge to define the concepts of ocean circulation and analyze the relationships between them, including spatiotemporal relationships and property relationships. Additionally, we will create semantic inference rules for ocean circulation.

### 3.2.1. Analysis of Semantic Concepts

The elements of the ocean circulation domain primarily involve the terminology around ocean currents, geographic locations, and temporal scales, as well as various geographical factors such as oceanic hydrology, meteorology, geological features, and environmental resources. Additionally, human factors, such as politics and economics, are also important. Achieving data interoperability directly can be challenging due to the heterogeneity of multi-source spatiotemporal data and differences in expressing relevant knowledge. Therefore, standardizing and converting data are necessary for effective sharing.

In this paper, the conceptual and relationship descriptions of ocean circulation are extracted as the foundation for constructing the ontology of ocean circulation. We aim to conceptualize the definitions related to ocean current phenomena by consulting resources such as standards [31–33], professional marine science books [34], academic papers [35], and the Internet [36,37]. This will help to establish a hierarchical structure for ocean circulation ontology. A class hierarchy can be constructed through top–down, bottom–up, or a combination of both approaches. The specific structure of the hierarchy is determined by the intended purpose and scope of the ontology. We will define the hierarchical structure of the semantic concepts of ocean circulation from aspects such as regional location, cause, scale, and properties, as shown in Figure 2.

Ocean circulation is a vast and intricate conceptual system. From different classification perspectives, various structures for ocean circulation ontology can be proposed. One approach is to categorize ocean currents based on thermal properties as warm, cold, or neutral. Alternatively, currents can be grouped regionally according to their distinctive characteristics, which are influenced by factors such as terrain, temperature, and salinity. This scheme yields categories such as equatorial, boundary, polar, and coastal currents. Another way to classify the ocean circulation system is by the factors that drive its formation and evolution, such as wind, temperature, salinity, Earth's rotation, and seabed topography. These factors interact to determine the formation and evolution of different types of currents, including wind-driven, density, slope, and compensation currents. In addition, the ocean circulation system is also affected by factors such as coastal terrain, wind forces above the sea surface, seawater density, and the Coriolis force, resulting in a variety of phenomena across different scales and sizes. Hence, there are different scales to consider in an ontology of ocean circulation, encompassing both spatial and temporal elements, including large-scale, mesoscale, and small-scale ones.

To construct an ontology for ocean circulation that meets semantic construction criteria, a conceptual hierarchical structure is established. This structure enhances the spatiotemporal reasoning and information retrieval capability of ocean circulation information. From a conceptual standpoint, this structure includes relationships between different concepts, relationships between concepts and attributes, and relationships between concepts and instances. The specific relationships involved are outlined in Table 1.

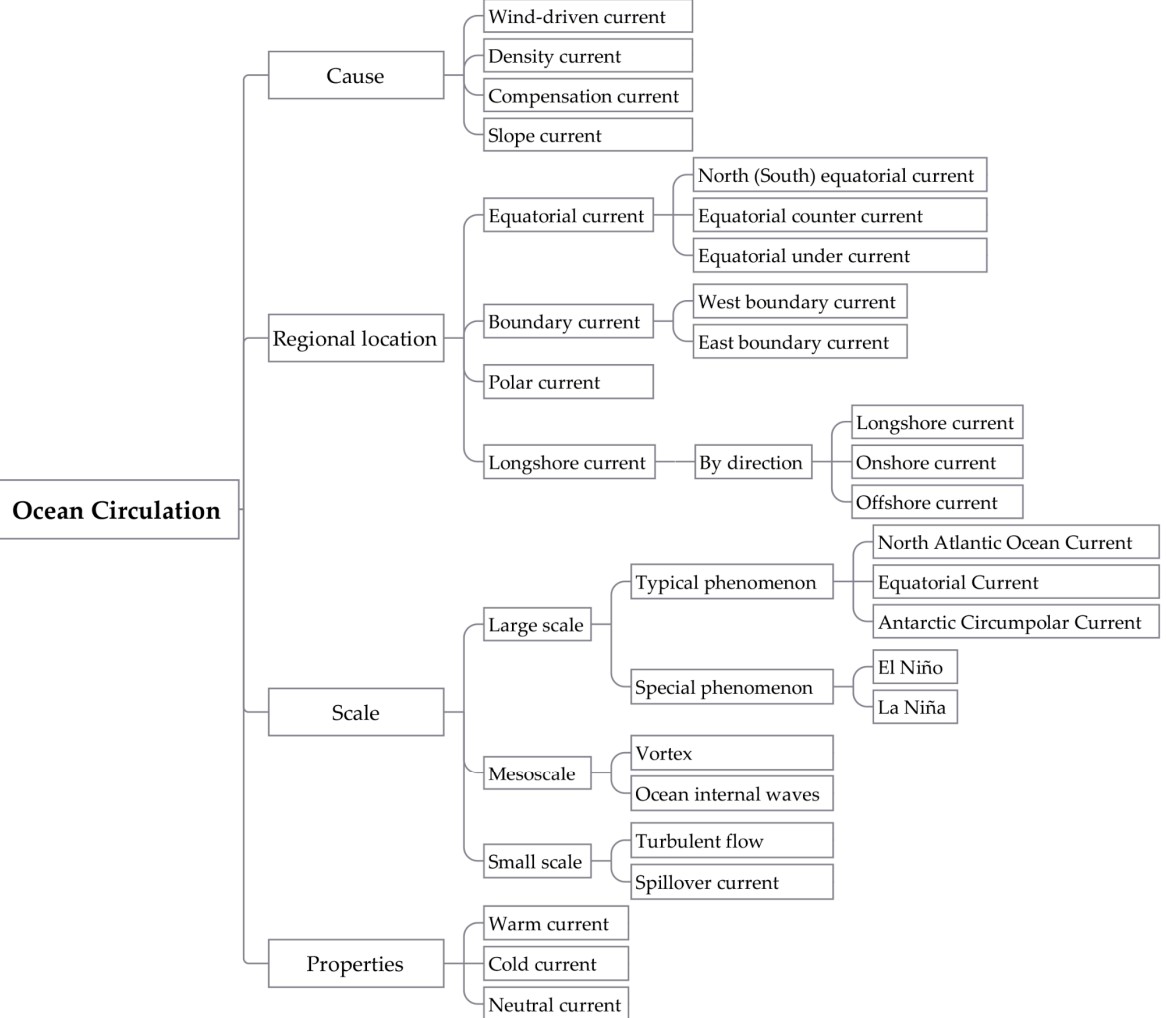

**Figure 2.** Building a hierarchical concept ontology of ocean circulation.

The basic conceptual relationships among ocean currents have been discussed above. Through an extensive review of the previous literature [38–42], five semantic relationships were selected: oc_exact, oc_subclass, oc_supclass, oc_null, and oc_overlap. The prefix 'oc_' indicates that the concept relates to ocean circulation. The first four relationships are relatively straightforward. The relationship of oc_overlap is used to indicate that two concepts are semantically related, but each concept has an attributed uniqueness that another concept does not have. To some extent, 'oc_overlap' indicates that two concepts are semantically related regardless of their degree of similarity [42].

Another crucial aspect of constructing an ontology for ocean circulation is expressing the multiple attributes of ocean currents. These attributes are multidimensional, encompassing basic motion features such as flow direction and speed, scalar field attributes such as temperature, pressure, and density, and semantic attributes related to time and space. This article categorizes ocean current attributes into basic, spatial, temporal, causal, functional, and predictability categories, which are illustrated in Figure 3.

**Table 1.** Semantic relationships between ocean circulation concepts.

| Relations | Symbols | Descriptions |
|---|---|---|
| Relationship between concepts. | equivalent-of | Concept equivalence, such as equatorial counter current and counter equatorial current. |
| | kinds-of | Whole-part relationship between concepts, such as ocean circulation and equatorial current. |
| | subclass-of | The relationship between a subclass and a superclass is that the subclass is a specific type of the superclass. For example, a western boundary current is a type of boundary current |
| Relationship between concepts and attributes. | attribute-of | Flow direction is an attribute of the concept of ocean current |
| Relationship between concepts and instances. | instance-of | The Brazil Current is an instance of the concept of warm current. |

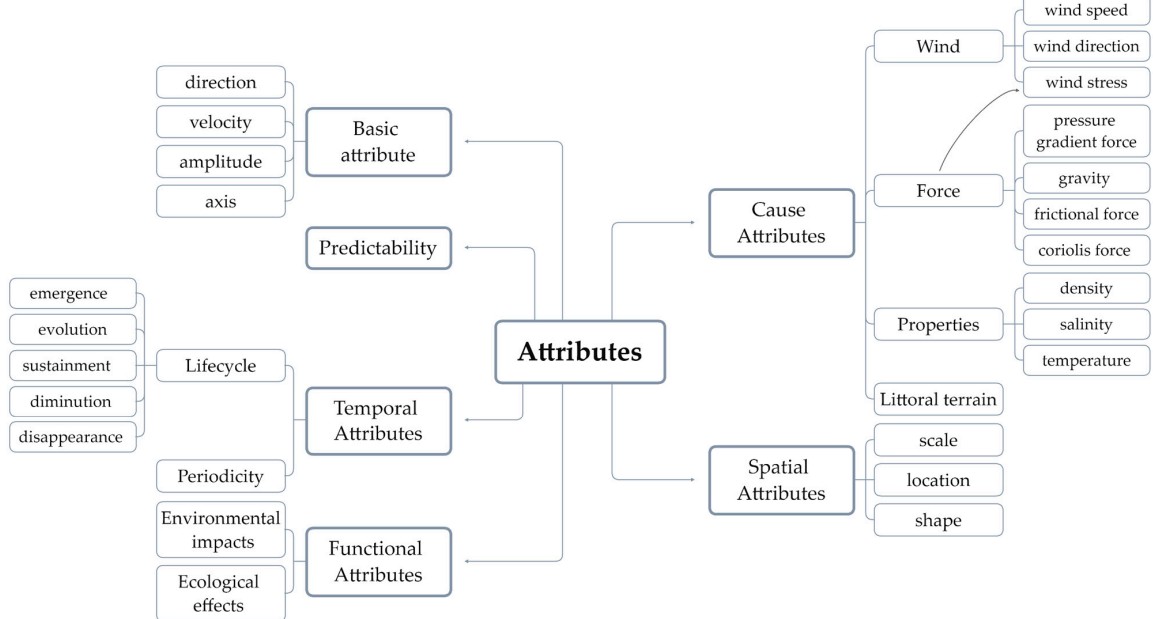

**Figure 3.** Semantic attributes of ocean circulation.

### 3.2.2. Analysis of Spatial Relationships

Providing a complete description of ocean currents requires the consideration of several key attributes, including their spatial position, shape, and scale. Spatial position refers to the geographic location of the current, which can be represented either qualitatively or quantitatively. Qualitative positions are broad and vague, while quantitative positions can be precisely represented on a map using the latitudinal and longitudinal coordinates. The spatial shape of ocean currents reflects their state within their environment and is typically two-dimensional. Spatial scale measures the size of the current and includes concepts such as length, width, height, area, and volume. These three spatial attributes are interconnected and reveal important information about the spatial characteristics of the current. The various spatial relationships of ocean circulation can be categorized into three parts: spatial distance relationships, spatial sequence relationships, and spatial topological relationships [43].

1.     Spatial distance relationships

Distance relationships express the relative position and closeness of spatial entities, which reflects the potential for interaction and exchange between neighboring objects in space. These relationships can be either qualitative or quantitative. Quantitative distance relationships typically disregard the target size and are calculated using Euclidean distance, as shown below.

$$D_{AB} = \sqrt{(x_A - x_B)^2 + (y_A - y_B)^2} \tag{2}$$

where $(x_A, y_A)$ and $(x_B, y_B)$ represent the coordinates of two objects in space.

When building an ontology, the threshold for metric relationships depends on the specific situation, which allows for conversions between qualitative and quantitative expressions. In this article, the threshold set is defined as $D\{D_i, D_{i+1}, D_{i+2}, \ldots\} \subseteq D$. The relationship between $D_s$ and $D_{AB}$ can be expressed as

$$D_s = \begin{cases} \text{very far} & D_{AB} \geq D_i \\ \text{far} & D_i < D_{AB} \leq D_{i+1} \\ \text{medium} & D_{i+1} < D_{AB} \leq D_{i+2} \\ \text{near} & D_{i+2} < D_{AB} \leq D_{i+3} \\ \ldots & \ldots \end{cases} \tag{3}$$

2.    Spatial sequence relationships

Sequential relationships, also known as positional or extensional relationships, define the relative orientation of geographical objects. In this article, the ocean circulation is treated as a planar entity, and only external positional relationships are taken into account, rather than expanding on specific internal positional relationships. The positional relationship of ocean circulation is established by assuming $P$ as the original target and $P_i$ as a point of target $P$. Similarly, $Q$ is the reference target and $Q_j$ is a point of Q. $\omega(P_i, Q_j)$ represents the angle between points $P_i$ and $Q_j$. $\theta(P_i, Q_j)$ represents the angle (north, south) deviating eastward. $\varphi(P_i, Q_j)$ represents the angle between the direction line and the plane. The sequence relationship is depicted in Table 2.

**Table 2.** Description of the spatial sequence relationships of ocean circulation.

| Direction | Description |
|---|---|
| East | $Restricted\_East(P_i, Q_j) = X(P_i) > X(Q_j) \ And \ Y(P_i) = Y(Q_j) \ And \ Z(P_i) = Z(Q_j)$ <br> $Restricted\_East(P_i, Q_j) = X(P_i) > X(Q_j) \ And \ Y(P_i) = Y(Q_j) \ And \ Z(P_i) \neq Z(Q_j)$ |
| South | $Restricted\_South(P_i, Q_j) = X(P_i) = X(Q_j) \ And \ Y(P_i) < Y(Q_j) \ And \ Z(P_i) = Z(Q_j)$ <br> $Restricted\_South(P_i, Q_j) = X(P_i) = X(Q_j) \ And \ Y(P_i) < Y(Q_j) \ And \ Z(P_i) \neq Z(Q_j)$ |
| West | $Restricted\_West(P_i, Q_j) = X(P_i) < X(Q_j) \ And \ Y(P_i) = Y(Q_j) \ And \ Z(P_i) = Z(Q_j)$ <br> $Restricted\_West(P_i, Q_j) = X(P_i) < X(Q_j) \ And \ Y(P_i) = Y(Q_j) \ And \ Z(P_i) \neq Z(Q_j)$ |
| North | $Restricted\_North(P_i, Q_j) = X(P_i) = X(Q_j) \ And \ Y(P_i) > Y(Q_j) \ And \ Z(P_i) = Z(Q_j)$ <br> $Restricted\_North(P_i, Q_j) = X(P_i) = X(Q_j) \ And \ Y(P_i) > Y(Q_j) \ And \ Z(P_i) \neq Z(Q_j)$ |
| Northwest | $North\_West(P_i, Q_j) = X(P_i) < X(Q_j) \ And \ Y(P_i) > Y(Q_j) \ And \ Z(P_i) = Z(Q_j)$ <br> $North\_West(P_i, Q_j) = X(P_i) < X(Q_j) \ And \ Y(P_i) > Y(Q_j) \quad Z(P_i) > Z(Q_j) \quad \varphi > 0°$ <br> $And \ Z(P_i) \neq Z(Q_j) \ And \ 90° < \theta > 180° \qquad Z(P_i) < Z(Q_j) \quad \varphi < 0°$ |
| Northeast | $North\_East(P_i, Q_j) = X(P_i) > X(Q_j) \ And \ Y(P_i) > Y(Q_j) \ And \ Z(P_i) = Z(Q_j)$ <br> $North\_East(P_i, Q_j) = X(P_i) > X(Q_j) \ And \ Y(P_i) > Y(Q_j) \quad Z(P_i) > Z(Q_j) \quad \varphi > 0°$ <br> $And \ Z(P_i) \neq Z(Q_j) \ And \ 0° < \theta > 90° \qquad Z(P_i) < Z(Q_j) \quad \varphi < 0°$ |
| Southwest | $South\_West(P_i, Q_j) = X(P_i) < X(Q_j) \ And \ Y(P_i) < Y(Q_j) \ And \ Z(P_i) = Z(Q_j)$ <br> $South_{West(P_i, Q_j)} = X(P_i) < X(Q_j) And \ Y(P_i) < Y(Q_j) \quad Z(P_i) > Z(Q_j) \quad \varphi > 0°$ <br> $And \ Z(P_i) \neq Z(Q_j) \ And \ 180° < \theta > 270° \qquad Z(P_i) < Z(Q_j) \quad \varphi < 0°$ |
| Southeast | $South\_East(P_i, Q_j) = X(P_i) > X(Q_j) \ And \ Y(P_i) < Y(Q_j) \ And \ Z(P_i) = Z(Q_j)$ <br> $South_{East(P_i, Q_j)} = X(P_i) > X(Q_j) And \ Y(P_i) < Y(Q_j) \quad Z(P_i) > Z(Q_j) \quad \varphi > 0°$ <br> $And \ Z(P_i) \neq Z(Q_j) \ And \ 270° < \theta > 360° \qquad Z(P_i) < Z(Q_j) \quad \varphi < 0°$ |
| Above | $Above(P_i, Q_j) = X(P_i) = X(Q_j) \ And \ Y(P_i) = Y(Q_j) \ And \ Z(P_i) > Z(Q_j)$ |
| Below | $Below(P_i, Q_j) = X(P_i) = X(Q_j) \ And \ Y(P_i) = Y(Q_j) \ And \ Z(P_i) < Z(Q_j)$ |

3.    Spatial topological relationships

Spatial topology is a qualitative spatial relationship that describes and analyzes the relative positions and relationships between geographical objects in space. This includes connectivity, adjacency, containment, intersection, nesting, mutual exclusion, and overlap [44]. The analysis of spatial topological structures helps to understand the organizational patterns and operational mechanisms of geographical space and is an important aspect of Geographic Information Systems (GIS). Established models for spatial topology include the RCC topological relationship model [45] and the N-intersection topological relationship model [46,47]. Based on point set topology, Egenhofer M.J.

and others developed the four-intersection and nine-intersection models [48]. Based on the nine-intersection model and basic topological structure, the expression of three-dimensional topological relationships is shown in Equation (4).

$$R_{\text{DE}-9\text{IM}} = \begin{bmatrix} \dim(A^0 \cap B^0) & \dim(A^0 \cap \partial B) & \dim(A^0 \cap B^-) \\ \dim(\partial A \cap B^0) & \dim(\partial A \cap \partial B) & \dim(\partial A \cap B^-) \\ \dim(A^- \cap B^0) & \dim(A^- \cap \partial B) & \dim(A^- \cap B^-) \end{bmatrix} \tag{4}$$

where $A^0$ and $B^0$ denote the internal points of their respective sets. The respective boundary point sets correspond to $\partial A$ and $\partial B$. $A^-$ and $B^-$ also represent the external point sets of their respective sets.

### 3.2.3. Analysis of Temporal Relationships

Time is an important dimensional feature in the spatiotemporal process of ocean circulation, describing the temporal process of the entire evolution of the flow phenomenon from formation to dissipation. Combined with the principles of time ontology [49], the temporal semantic expression of ocean circulation is generally described through time points and time relations, and time points are usually expressed with the help of the concept of time granularity [50]. Time granularity specifies the precision level in time descriptions. Depending on the duration of an event, it can be divided into units such as years, months, days, hours, minutes, or seconds. Alternatively, it can be divided into seasonal (spring, summer, autumn, winter) or solar divisions. When considering temporal relationships in the context of current lifecycles, it is important to account for both time points and periods. Three types of relationships can be established based on the occurrence times of different current instances. These relationships include earlier than, later than, and equal to. For example, current 'A' may occur earlier than current 'B'. Temporal relationships can be expressed in various ways, such as Allen's interval algebra theory [51]. This theory allows for the derivation of 13 temporal relationships by comparing the relationships between the endpoints of two time periods. Provided events $T_1$ and $T_2$, where $t_b^1$ and $t_e^1$, and $t_b^2$ and $t_e^2$ denote the commencement and termination of $T_1$ and $T_2$ respectively, the temporal relationships in ocean circulation can be conceptualized as illustrated in Table 3.

**Table 3.** The temporal relationships of ocean circulation.

| Relationships | Definition | Descriptions |
|---|---|---|
| Earlier than | $t_e^1 < t_b^2$ | $T_1$'s end time is earlier than $T_2$'s start time. |
| Later than | $t_b^1 > t_e^2$ | $T_1$'s start time is later than $T_2$'s end time |
| Equal to | $(t_b^1 = t_b^2) \cap (t_e^1 = t_e^2)$ | $T_1$ and $T_2$ have equal start and end times |
| Overlap | $(t_b^1 < t_b^2) \cup (t_b^2 < t_e^1 < t_e^2)$ | Part of $T_1$ is contained within $T_2$ |
| Meet | $t_e^1 = t_b^2$ | $T_1$'s end time is equal to $T_2$'s start time |
| Contain | $(t_b^1 < t_b^2) \cap (t_e^1 > t_e^2)$ | $T_2$ is contained within $T_1$ |

### 3.2.4. Semantic Inference Analysis of Ocean Circulation

To enhance data retrieval efficiency by accessing implicit knowledge within an ontology, it is typical to utilize knowledge rules that are based on the ontology knowledge system or the expression of ontological concepts and relationships. This enables semantic reasoning within ontology to discover implicit information. The former describes ontological relationships to achieve semantic reasoning, while the latter formulates knowledge rules to expand ontological concepts and discover more implicit knowledge. Building upon these methods, we establish semantic inference rules and expression methods for ocean circulation by analyzing the semantic relationships within the spatiotemporal development of ocean circulation. In the following subsections, we will discuss the semantic inference rules for ocean circulation from three perspectives: conceptual relationships, spatial relationships, and temporal relationships.

1. Inference analysis of concept relationships

In Section 3.1, we examined the interrelationships among various concepts related to ocean circulation. Thereafter, the semantic relationships of concepts in a tree-based ontology could be readily inferred based on the hierarchical structure of ocean current concepts that we have proposed (such as inheritance property of parent–child concepts). This hierarchical structure can be further expanded to infer semantic relationships between concepts from different ontologies. In the subsequent discussion, 'A' and 'B' denote different concepts in ocean circulation ontology.

(1)    If the semantic relationship between concepts A and B is oc_exact, these two concepts have the same meaning, and concept A shares the same semantic relationships with concept B.

(2)    If the semantic relationship between concepts A and B is oc_subclass, concept A is a special type of concept B.

(3)    If the semantic relationship between concepts A and B is oc_supclass, the semantic relationship between concept A's parents and concept B is oc_supclass.

(4)    If the semantic relationship between concepts A and B is sem_null, then no new relationship can be inferred.

Therefore, the inference rules can be expressed as follows.

$$f(A,B) = \begin{cases} oc\_subclass(A, B's\ parent)\ if\ oc\_exat(A,B) = True\ or\ oc\_subclass(A,B) = True \\ oc\_supclass(A's\ parent, B)\ if\ oc\_exat(A,B) = True\ or\ oc\_supclass(A,B) = True \end{cases} \quad (5)$$

2.    Inference analysis of spatial relationships

The determination of spatial semantic relationships in ocean circulation can take into account both spatial distance and spatial sequence relationships. We assume that 'A', 'B', and 'C' are three objects within the ocean circulation domain, with their respective distances defined as $d_1$, $d_2$, and $d_3$. From this point, we can draw the following conclusions shown in Equation (6).

$$f_{\text{distance}}(A,B,C) = \begin{cases} d_1 = d_3\ if\ d_1 = d_2\ and\ d_2 = d_3 \\ d_1 < d_3\ if\ d_1 < d_2\ and\ d_2 < d_3 \\ d_1 > d_3\ if\ d_1 > d_2\ and\ d_2 > d_3 \\ d_2 < d_3\ if\ d_1 > d_2\ and\ d_1 < d_3 \\ d_2 > d_3\ if\ d_1 < d_2\ and\ d_1 > d_3 \\ d_1 < d_2\ if\ d_3 > d_1\ and\ d_3 < d_2 \\ d_1 > d_2\ if\ d_3 < d_1\ and\ d_3 > d_2 \end{cases} \quad (6)$$

As for the spatial sequence relationships, d above B and B is located above C, then we can infer that A is located above C.

3.    Inference analysis of temporal relationships

Time entities in ocean circulation include time point objects and time period objects. Regarding time point entities, time point objects $t_p^1$, $t_p^2$, and $t_p^3$ are introduced, and then the following relationships can be derived.

$$f_{\text{time\_point}}(t_p^1, t_p^2, t_p^3) = \begin{cases} t_p^1 = t_p^3\ if\ t_p^1 = t_p^2\ and\ t_p^2 = t_p^3 \\ t_p^1 < t_p^3\ if\ t_p^1 < t_p^2\ and\ t_p^2 < t_p^3 \\ t_p^1 > t_p^3\ if\ t_p^1 > t_p^2\ and\ t_p^2 > t_p^3 \\ t_p^2 < t_p^3\ if\ t_p^1 > t_p^2\ and\ t_p^1 < t_p^3 \\ t_p^2 > t_p^3\ if\ t_p^1 < t_p^2\ and\ t_p^1 > t_p^3 \\ t_p^1 < t_p^2\ if\ t_p^3 > t_p^1\ and\ t_p^3 < t_p^2 \\ t_p^1 > t_p^2\ if\ t_p^3 < t_p^1\ and\ t_p^3 > t_p^2 \end{cases} \quad (7)$$

As for time period entities, temporal semantic relationships are typically determined by examining the start and end times of time objects. Based on previous analyses of temporal semantic relationships, a time period object $T_3$ is introduced, with start and end times denoted as $t_b^3$ and $t_e^3$, respectively. This allows the derivation of the following temporal semantic reasoning, and the inference rules can be expressed as follows:

$$f_{\text{time\_period}}(T_1, T_2, T_3) = \begin{cases} T_1 = T_2 = T_3\ if\ t_b^1 = t_b^2 = t_b^3\ and\ t_e^1 = t_e^2 = t_e^3 \\ T_1 < T_3\ if\ t_e^1 \leq t_b^2\ and\ t_e^2 \leq t_b^3\ and\ t_b^3 \neq t_e^2 \\ T_1 > T_3\ if\ t_b^1 \geq t_e^2\ and\ t_b^2 \geq t_e^3\ and\ t_b^2 \neq t_e^2 \\ T_2 < T_3\ if\ t_b^1 \geq t_e^2\ and\ t_e^1 \leq t_b^3\ and\ t_b^1 \neq t_e^1 \\ T_2 > T_3\ if\ t_e^1 \leq t_b^2\ and\ t_b^1 \geq t_e^3\ and\ t_b^1 \neq t_e^1 \\ T_1 < T_2\ if\ t_b^3 \geq t_e^1\ and\ t_e^3 \leq t_b^2\ and\ t_b^3 \neq t_e^3 \\ T_1 > T_2\ if\ t_e^3 \leq t_b^1\ and\ t_b^3 \geq t_e^2\ and\ t_b^3 \neq t_e^3 \end{cases} \quad (8)$$

### 3.3. Modeling and Formal Expression of Ocean Circulation Domain Ontology

Ontology description languages employ specific formalized representation methods with which to express ontology models within a domain. OWL, as one of the most widely used ontology modeling languages, offers unique advantages in building ontologies. Therefore, we utilized OWL to

construct an ontology for ocean circulation based on concepts, attributes, and semantic relationships. Due to the complexity of ocean circulation involving many spatiotemporal concepts, the following chart diagram will simplify the formal expression of ocean circulation. We will outline the formal construction process of ocean currents in several steps.

Step 1: Define the classes and the class hierarchy.

It is crucial to clearly define key terms related to ocean circulation. Based on the analysis of the semantics of ocean circulation concepts in Section 3.1, we reused existing concepts from established ontologies, such as SWEET, to represent the knowledge of oceanography and try to maintain consistency in these concepts within the class hierarchy. By managing this, we can reduce the workload of constructing our ontology and provide support for interoperability among ontologies within multiple domains. Here, we provide an example of ocean currents, as illustrated in Figure 4.

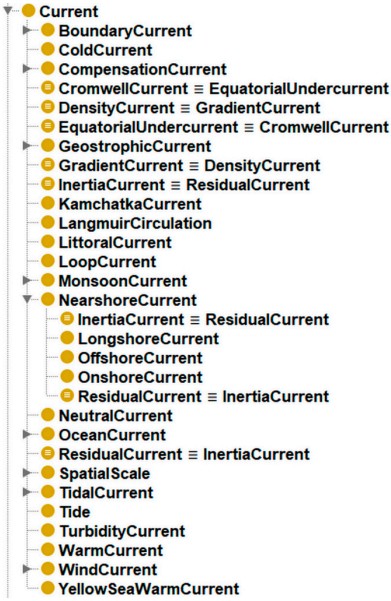

**Figure 4.** Class hierarchy for ocean currents.

Step 2: Define the class properties.

In OWL, it is necessary to define both object properties and data properties for an ocean circulation ontology. In this step, we also refer to the definitions of the relevant property elements within the SWEET ontology. Object properties are used to describe the relationships between two entities. They represent associations, connections, or relationships between different individuals, while data properties are used to describe the characteristics, attributes, or individual values. They are usually linked to specific data types, such as strings, integers, dates, etc. An object property is commonly defined as an instance of the built-in OWL class owl: topObjectProperty, as illustrated in Figure 5a. Similarly, a data property is typically defined as an instance of the built-in OWL class owl: topDataProperty, as shown in Figure 5b.

Step 3: Define the restrictions associated with a class directly.

In this step, we have established constraints for constructing an ontology of ocean circulation. For instance, we have defined restrictions by which to specify the domain and range of concepts. We assert that the spatial distance value cannot be negative. The data property "time" should be filtered by "date TimeStamp", and the constraint should be defined as "Some (existential)".

Step 4: Create instances.

Based on the definition of classes and properties, some instances of ocean currents were created, such as El Niño, Norwegian current, and Gulf stream.

Through the aforementioned formal representation of concepts and attribute relationships, we have constructed an ontology for ocean circulation. examples of the ontology can be seen in Figures 6 and 7.

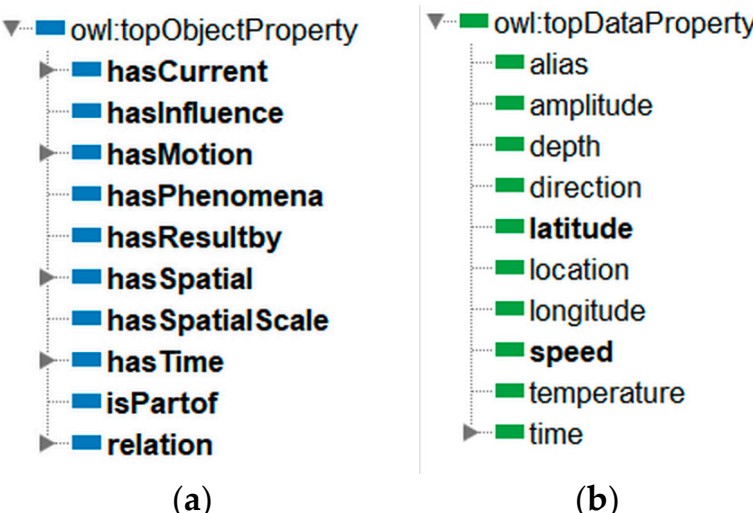

**Figure 5.** Definition of the class properties: (**a**) object properties to represent extracted attributes, and (**b**) data properties to represent extracted attributes.

**Figure 6.** Concept relationships of ocean currents: the graph generated by the OntoGraf tab in Protégé OWL.

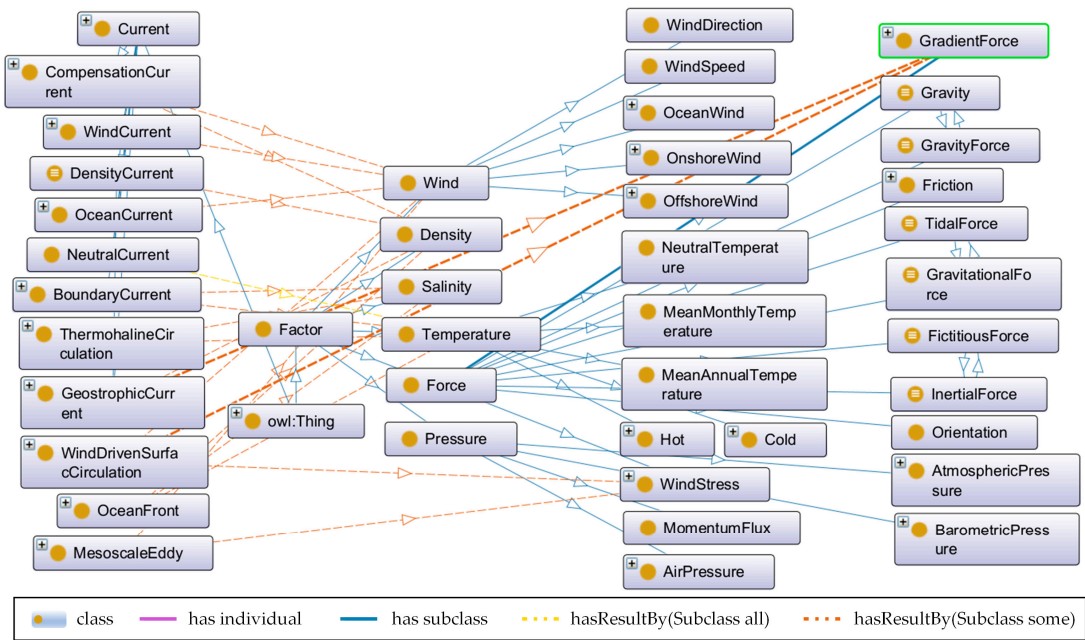

**Figure 7.** The ontological expression of cause attributes in ocean circulation: graph generated by the OntoGraf tab in Protégé OWL.

Figure 6 shows the class hierarchy of ocean currents. The OntoGraf tab of Protégé is used to develop this graph. Each block in the figure represents a class or an individual, with each class node representing a specific concept or a related concept. Solid lines represent subclass relationships (i.e., "has subclass") or instance relationships (i.e., "has individual"). This hierarchical classification allows us to gain a comprehensive understanding of the inherent relationships between ocean currents and to infer specific information about them from higher-level categories.

Figure 7 depicts the ontological expression of cause attributes in ocean circulation. From the hierarchical relationships of concepts, we can understand that the factors affecting ocean circulation include wind, temperature, salinity, density, and force, as well as determine which factors impact specific ocean currents. Based on their intrinsic connections, we can quickly locate and infer the implicit knowledge we need.

*3.4. Ontology Application Experiment*

In order to validate the aforementioned ontology model, we took the Equatorial Counter Current as an example to illustrate the formal description of ocean circulation.

The Equatorial Counter Current, also known as the North Equatorial Countercur-rent (NECC), flows from west to east to compensate for the ocean water carried away by the equatorial current in the eastern part of the ocean, which exhibits compensatory and inclined current characteristics. The Equatorial Counter Current is located between 3 and 5 degrees north latitude and 10–12 degrees north latitude, with flow rates generally ranging from 40 to 60 cm/s and reaching a maximum of 150 cm/s. In winter, flow rates decrease to 15–30 cm/s or less.

Figure 8 displays the dynamics of the Equatorial Counter Current on 2 May. It can be seen that the fuzzy boundary characteristics of the Equatorial Counter Current and its multi-dimensional dynamic attributes, especially the change in the flow direction, which flows predominantly from west to east. The Ocean circulation takes on various forms across different temporal and spatial scales, while remaining interconnected. For example, the Equatorial Counter Current, as well as the North and South Equatorial Currents, make up a complex circulation system in the tropics. Seawater converges from all directions in some areas, which forces the surface water down and creates a downward current. In other areas, the seawater diverges to allow the deep water to rise and replenish the surface, ultimately creating an upward current. Through this process, nutrients from the lower layers reach the surface, which promotes the growth of plankton and deepens and decreases the transparency of the seawater.

From the above description, combined with previous analysis of ocean circulation semantics, the following ontological structure of the Equatorial Counter Current can be extracted. Figure 9 illustrates a visualization of the Equatorial Counter Current's ontology.

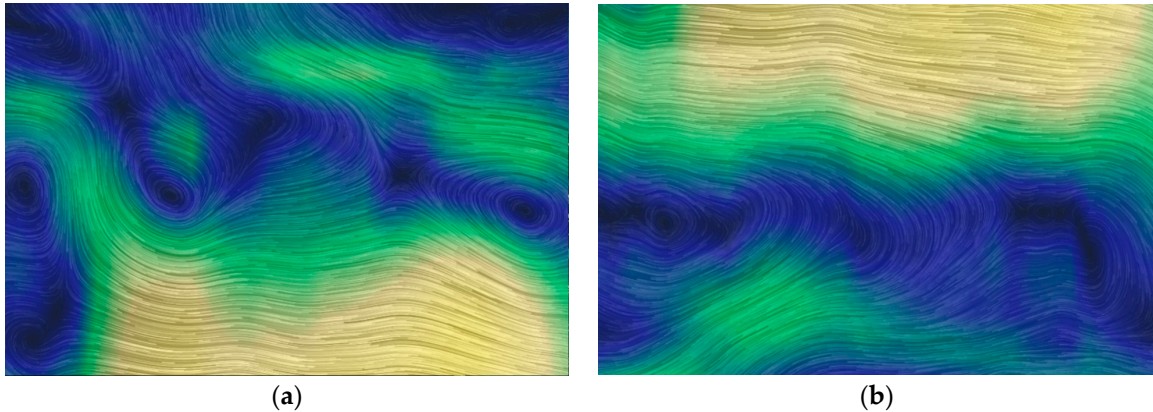

(a)                                        (b)

**Figure 8.** The symbolization of the Equatorial Counter Current: (**a**) the symbolization of the North Equatorial Counter Current and (**b**) the symbolization of the South Equatorial Counter Current.

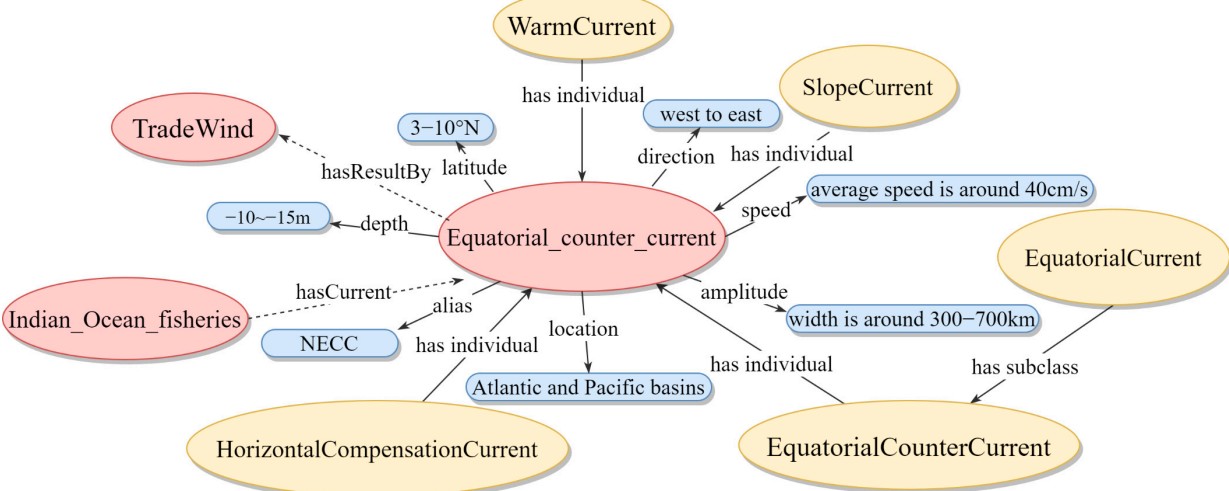

**Figure 9.** The ontological visualization of the Equatorial Counter Current.

$O_{Equatorial\_Counter\_Current}$ = {C(Current, Ocean_Current, Equatorial _Current, Equatorial_Counter_ Current, Horizontal_Compensation_Current, Slope_Current, Warm_Current), R((subclass of Equatorial _Current), (disjointWith Polar_Current)), P((Atlantic and Pacific basins), (average speed is around 40 cm/s), (−10~−15 m), (3–10° N), (width is around 300–700 km), (west to east)), I(Equatorial Counter Current)}.

In addition, based on the previously logical reasoning rules of ocean circulation, in this subsection, we use the HermiT reasoner built into Protégé to check the reasoning effects of certain instances.

- Semantic relationship expression of instances.

Here, we take the Indian Ocean fishing grounds as an example for verification. The following is a brief description of the elements of the Indian Ocean fishing grounds.

"In the northwestern region of the Indian Ocean, particularly in the Arabian Sea. The influence of the Somali Current and the Equatorial Counter Current results in high salinity and elevated water temperatures. These conditions contribute to the presence of extensive upwellings that make it become an optimal environment to form cephalopod fishing grounds".

From this information, it can be inferred that, due to the influence of the Somali Current and Equatorial Counter Current, the Indian Ocean fisheries in this area represent an instance of a mixed fishery. The partial code for describing the conceptual relationship of the Indian Ocean fishing ground instance is shown below. Figure 10 vividly illustrates the semantic relationship of this component.

*Define* Class: Firsheries, JoinFishingGround, SomaliCurrent, EquatorialCounterCurrent, ColdCurrent, WarmCurrent
*Define* Instance: Indian_Ocean_fisheries, SomaliCurrent, Equatorial_counter_current, TradeWind
*Define* Object property: hasCurrent, hasResultby
*Define* Data property: depth, alias, speed, location, amplitude, latitude, direction

Indian_Ocean_fisheries *instance of* Class Firsheries
Indian_Ocean_fisheries hasCurrent SomaliCurrent
Indian_Ocean_fisheries hasCurrent Equatorial_counter_current
SomaliCurrent *instance of* SomaliCurrent
Equatorial_counter_current depth "-10~15m"
Equatorial_counter_current alias "North Equatorial Counter Current (NECC)"
Equatorial_counter_current speed "average speed is around 40cm/s"
Equatorial_counter_current location "Atlantic and Pacific basins"
Equatorial_counter_current amplitude "width is around 300~700km"
Equatorial_counter_current latitude "3-10°N"
Equatorial_counter_current direction "west to east"
JoinFishingGround equivalent to (hasCurrent *some* EquatorialCounterCurrent) *and* (hasCurrent *some* SomaliCurrent)
JoinFishingGround equivalent to (hasCurrent *some* ColdCurrent) and (hasCurrent *some* WarmCurrent)

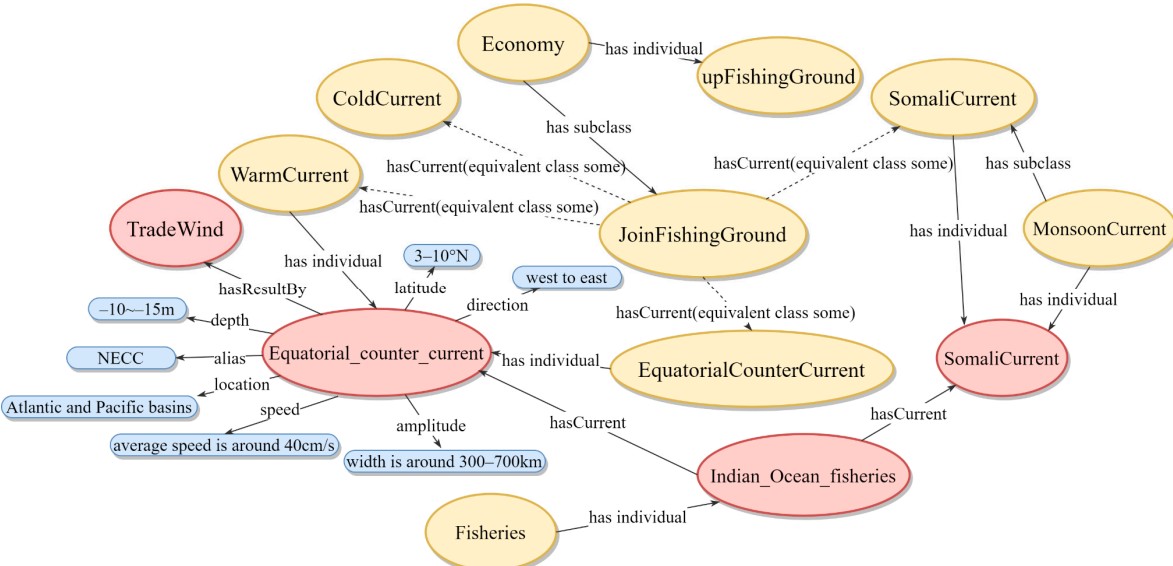

**Figure 10.** Description of the semantic reasoning relationship for the Indian Ocean fisheries.

- Validate the reasoning results.

Based on the description of the semantic relationships of the instance mentioned, we obtained the following reasoning results using HermiT reasoner (In Figure 11, the results inferred are represented by the yellow section).

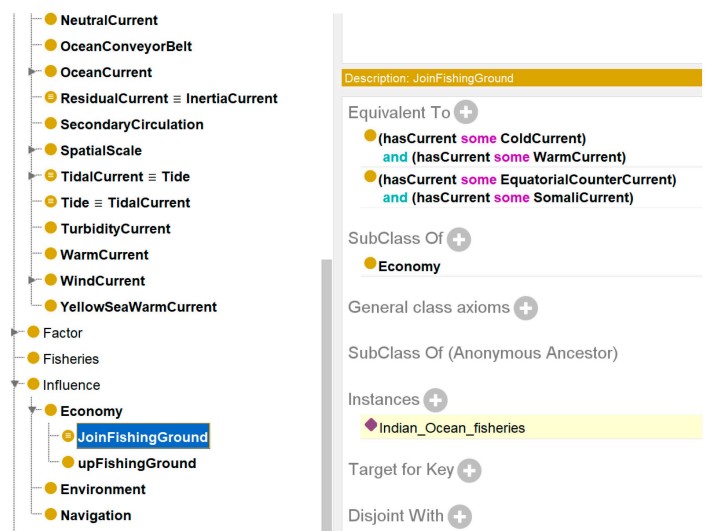

**Figure 11.** Reasoning results of the instance.

## 4. Ontology-Based Ocean Circulation Knowledge Base

The construction of an ontology for ocean circulation has created a unified representation of concepts and their associated elements at a semantic level. The standardized processing of multi-source spatiotemporal data has enabled a unified description framework for all such data. This ensures consistency in the handling of spatiotemporal data. We have established a thematic database for ocean currents, with the goal of standardizing the representation of information within this field.

### 4.1. The Framework of Ocean Circulation Knowledge Base

This paper proposes an ontology-based framework for the ocean circulation system; our vision is to present a novel approach to improve the complete expression of the ocean circulation knowledge system, which provides a unified description for organizing data, analyzing spatial–temporal data, and performing data sharing. For this purpose, the authors suggest that the framework be divided into three main layers: the domain layer, the service layer, and the application layer (Figure 12). Each layer is organically connected to the next layer through interfaces and functions, thus ensuring the modular structure and functional division of the overall system.

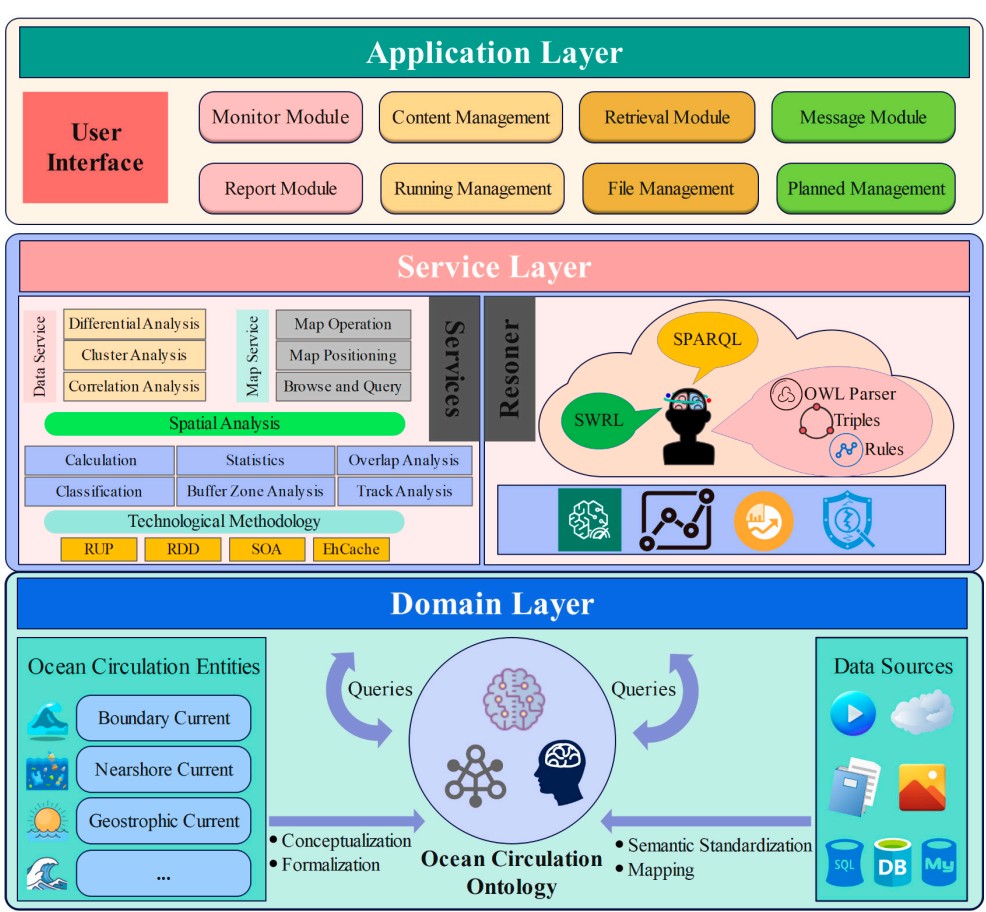

**Figure 12.** The framework of the ontology-based knowledge base of ocean circulation.

The domain layer (the bottom part of Figure 12) is the central component of the ocean circulation knowledge base, which offers a formal description of the ontological knowledge related to ocean circulation. This defines the ontology model of concepts, including semantic relationships such as parent–child, equivalence, and mutual exclusion. It also includes spatial–temporal semantic relationships, such as spatial and temporal relationships. There are distinct data sources available that are included in a domain layer to provide the data amount, data integration, and data performance from queries on the ontology. This knowledge base is considered to perform data integration with distinct data sources in a unified database. This layer communicates with the service layer to exchange information. By utilizing ontological reasoning, the correlation between the main classes of ontology could provide implicit information with which to satisfy user expectations. Technical methods are

used in the framework to logically analyze and reason about the feedback results from the domain layer, which provides the top layer with accurate and appropriate application services.

The service layer (the middle part of Figure 12) utilizes the technical methodology to realize data service, spatial analysis, and map service based on the domain layer using SPARQL Protocol and RDF Query Language (SPARQL) query specification and Semantic Web Rule Language (SWRL) inference rules. The service layer is further divided into two sub-layers: the technical service layer and the reasoning layer. The reasoning layer (the right side of the middle part in Figure 12) can effectively infer flow field information based on semantic relationship descriptions and established semantic rules. This reasoning layer includes components such as SPARQL, SWRL, and an inference engine. The technical service layer (the left side of the middle part in Figure 12) describes the technical infrastructure that supports the construction of the knowledge base. This part typically includes software technologies involved in the system development process, such as Rational Unified Process (RUP), Service-Oriented Architectures (SOA), Responsibility-Driven Design (RDD), and others. Additionally, the technical service layer provides basic data analysis based on the ontology library and achieves spatial analysis functions such as spatial information calculation, classification, and overlay analysis. It combines semantic relationships to implement map service. The service layer provides technological service support, which plays a crucial role in ensuring the stability of human–computer interaction for the upper application layer.

Many marine services require integrated marine data management and efficient data retrieval. The application layer (the top part of Figure 12) addresses these requirements by providing a unified and concise portal to the ontological knowledge base. This effectively displays thematic information related to ocean circulation. Our goal is to provide scientific research personnel and the public with data services that facilitate the organized and efficient use of marine data, while also promoting the sharing of marine information.

### 4.2. Ocean Circulation Ontology Query Example

We utilize the SPARQL query language to extract knowledge from the ocean circulation knowledge base. In this case, we focus on large-scale ocean current phenomena as an example. The search criteria aim to retrieve large-scale ocean circulation phenomena influenced by wind and possessing attributes such as temperature (Tp), location (Lp), and time (Pp). The SPARQL description is as follows:

```
1   Select ?phenomenon ?Tp ?Lp ?Pp
2   where {
3              ?phenomenon rdf:type oc:LargeScaleOceanCurrent.
4              ?phenomenon oc:hasResultby oc:TradeWind.
5              ?phenomenon oc:temperature ?Tp.
6              ?phenomenon oc:location ?Lp.
7              ?phenomenon oc:period ?Pp.
8   }
```

Table 4 presents the retrieved results. By applying the defined search criteria, we have identified two distinct oceanic phenomena: El Niño and La Niña phenomena. This result effectively demonstrates the semantic connections between the concepts of ocean circulation and confirms the effectiveness of our description of these relationships within ocean circulation.

**Table 4.** Retrieval results from the ontological knowledge base.

| Phenomenon | Tp | Pp | Lp |
|---|---|---|---|
| El Niño | rise abnormally | between two and seven years | in the central and east-central equatorial Pacific |
| La Niña | reduce abnormally | every few years | the eastern equatorial part of the central Pacific Ocean |

## 5. Discussion and Conclusions

Ocean circulation plays a fundamental role in oceanographic research. With continuous advancements in ocean observation techniques, there has been a substantial growth in ocean circulation data. However, the diversity in data sources and representation has resulted in semantic heterogeneity, which poses challenges for effective data utilization and sharing. Bridging the semantic gap has emerged as a prominent focus of interest among scholars in this field. In order to address this problem, this paper introduced the relevant theories and methods of ontology and proposed a quintuple-based ontological structure for ocean circulation after careful consideration of the knowledge system and relational characteristics of ocean circulation. This structure provides a normative framework for organizing ocean circulation knowledge. This study further performed the semantic analysis and

definition of concepts, attributes, spatiotemporal relationships, and reasoning, and then provided a formal description and expression of the domain ontology of ocean circulation with OWL language. Furthermore, an ontological knowledge base of ocean circulation was introduced based on a formal description of ocean circulation. This knowledge base employs a three-layer architecture in order to achieve an integrated process that includes ontology construction, technical services, inference, and system application management. This architecture aims to effectively describe and share the knowledge related to ocean circulation.

This article presented a formal expression of ocean circulation based on the division rules and spatiotemporal characteristics of ocean circulation. We used the descriptions of oceanographic elements from existing ontologies to streamline the process of developing our own ontology. Due to the dynamic nature of ocean currents, the descriptions of knowledge related to ocean circulation in existing ontologies were insufficient for our requirements. As a result, we had to redefine certain concepts and intrinsic relationships in order to fully construct the ontology of ocean circulation. In the future, there should be more detailed descriptions of the complex spatiotemporal relationships involved in ocean circulation dynamics. This can be achieved by referencing additional data types and expanding the OWL modeling primitives of the ocean circulation ontology, thereby improving the overall representation of the ontology. At the same time, we also need to consider ontology alignment. Although we have made progress in identifying links between concepts and property relationships in the reuse of existing ontological knowledge, our work is not yet perfect. We need to continue exploring the alignment of conceptual hierarchical representations in order to improve our approach in the next phase.

The construction of an ontological knowledge base provides an effective means for knowledge sharing and integration in ocean circulation. However, it is essentially a massive project that involves a vast amount of domain-specific knowledge and requires the participation of domain experts. The ontological knowledge base is predominantly constructed manually, which can result in lower efficiency. Future work should focus on improving the fine-grained description of the ontological knowledge base and enhancing the efficiency of ontology construction. Furthermore, the validation of an ontology is crucial, as it directly affects the accuracy of data description and the scalability of the ontology.

In summary, this study presented a preliminary investigation of the semantic expression of the spatiotemporal characteristics of ocean circulation. To some extent, it provides a framework for analyzing ocean circulation, with massive data collected from numerous different resources, which will provide theoretical and methodological support for extracting and mining implicit semantic relationships within ocean circulation.

**Author Contributions:** Conceptualization, H.Z., A.Z., C.W. and L.Z.; methodology, H.Z., C.W. and S.L.; formal analysis, H.Z.; writing—original draft preparation, H.Z.; writing—review and editing, A.Z., L.Z. and C.W.; visualization, H.Z.; supervision, A.Z.; funding acquisition, A.Z. All authors have read and agreed to the published version of the manuscript.

**Funding:** This project was partially supported by the Key Program of Marine Economy Development, Special Foundation of the Department of Natural Resources of Guangdong Province, China Project no. (GDNRC [2022]19).

**Institutional Review Board Statement:** Not applicable.

**Informed Consent Statement:** Not applicable.

**Data Availability Statement:** Not applicable.

**Acknowledgments:** We extend our gratitude to colleagues in the Marine Technology Laboratory for their assistance in conceptualization. We also express our appreciation to the marine experts for their guidance and support.

**Conflicts of Interest:** The authors declare no conflict of interest.

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
