# Peer review of "Research on Construction and Application of Ocean Circulation Spatial–Temporal Ontology"

_jmse, doi:10.3390/jmse11061252_

Round 1

Reviewer 1 Report

The manuscript is devoted to the space-time ontology of ocean circulation. The authors focus on the analysis of temporal and spatial relationships of ocean circulation. Based on the ontological description of ocean circulation, a specialized knowledge base structure was developed. This is a good manuscript, but there are a number of comments to it:

1. The abstract is not written correctly, it is necessary to write in it what exactly has been done, what the manuscript is about without introductory words.

2. The inscriptions inside the drawings are very small, it is desirable to enlarge.

3. It is better to arrange Table 2 differently, so it merges and it is not clear which line begins where.

4. The article talks about data, but does not specify which data will be used in this model.

5. The introduction describes many models and methods, but there is no advantage of this model or indication of the disadvantages of the described models

There are temporary inaccuracies in some proposals

Author Response

Thank you for your valuable feedback and insightful comments on our paper. We have carefully considered each of your suggestions and have made the necessary revisions to improve the quality and clarity of our research.

Please refer to the attached document for a detailed response to each comments and the corresponding modifications we have made in the revised manuscript. We have addressed the concerns and suggestions raised by the reviewers and believe that these changes have significantly improved the quality and contribution of our research.

Once again, we would like to express our gratitude for your thorough review and valuable insights. Your comments have played a crucial role in shaping our research and strengthening our paper.

Thank you for your time and consideration.

Reviewer 2 Report

- The paper is very lengthy and hard to follow. The work that was done seems important and relevant, but it is very hard to assess its novelty and marginal contribution. The related work is cited and described in length, but the specific contribution of this paper is not described with respect to the related work. Thus one cannot understand what is this paper's novelty and, why the related work is different or how exactly it relates to your work. 

- There are some problems with the exposition in Section 3. In multiple places, there are paragraphs that seem to belong in the introduction repeating themes of why ontologies are important or other motivational topics. Furthermore, there are several claims that are made without justification or citation. The most glaring one is your claim that you improve upon the ontology construction method from Noy et al. but without explaining how your method differs and justifying why it would be an improvement. 

- There is a wealth of existing ontologies in the environmental/oceanographic domain. It is unclear why you would redefine concepts from the more general oceanographic domain, such as climate models and measured variables when these are already well defined. This decision needs to be justified or at least these parallel concepts need to be aligned with the external concepts to allow interoperability of your ontology with other ontologies. 

- The ontology itself and the other assets described are not published as open source or attached to the submission; this prohibits reuse, and limits their evaluation and contribution to science. 

- It is unclear if the root of the ontology and the core concepts are aligned with base ontologies such as the OBO core ontology or relation ontology. Without these, again, it's hard to interoperate this ontology with others. 

The level of English is shaky, with many missing prepositions, bad grammar, and sometimes weird use of terms. I attached some comments in the PDF, but someone that is a native English speaker needs to proofread this before publication. 

Author Response

Thank you for your thorough review of our paper and for providing us with valuable feedback and suggestions. We sincerely appreciate the time and effort you have dedicated to evaluating our work.

We have carefully reviewed each of your comments and made the necessary revisions to address the concerns raised. Please find attached a detailed response to each comment, which outlines the modifications we have made in the revised manuscript.

We have reviewed your feedback and believe that the revisions we made have enhanced the clarity, coherence, and overall quality of our research. We have provided explanations and justifications for the decisions made in response to your comments.

Once again, we would like to express our sincere gratitude for your helpful feedback and valuable insights. Your expertise and insights have been instrumental in improving our paper.

Thank you for your time and consideration.

Round 2

Reviewer 2 Report

The authors have made a serious effort to address my comments, and the revised manuscript is profoundly more readable and showcases their contribution in a way that does it justice. 
The English language level has improved as well. I thank the authors for choosing to invest the time and effort required in aligning their ontology with other marine ontologies and sharing their work for the benefit of the community. 
I wish to congratulate the authors on a job well done.